# Magnesium oxide-water compounds at megabar pressure and implications on planetary interiors

Shuning Pan[1,5], Tianheng Huang[1,5], Allona Vazan [2], Zhixin Liang[1], Cong Liu [1], Junjie Wang[1], Chris J. Pickard [3,4], Hui-Tian Wang [1], Dingyu Xing[1] & Jian Sun [1] ✉

Magnesium Oxide (MgO) and water ($H_2O$) are abundant in the interior of planets. Their properties, and in particular their interaction, significantly affect the planet interior structure and thermal evolution. Here, using crystal structure predictions and ab initio molecular dynamics simulations, we find that MgO and $H_2O$ can react again at ultrahigh pressure, although $Mg(OH)_2$ decomposes at low pressure. The reemergent $MgO \cdot H_2O$ compounds are: $Mg_2O_3H_2$ above 400 GPa, $MgO_3H_4$ above 600 GPa, and $MgO_4H_6$ in the pressure range of 270–600 GPa. Importantly, $MgO_4H_6$ contains 57.3 wt % of water, which is a much higher water content than any reported hydrous mineral. Our results suggest that a substantial amount of water can be stored in MgO rock in the deep interiors of Earth to Neptune mass planets. Based on molecular dynamics simulations we show that these three compounds exhibit superionic behavior at the pressure-temperature conditions as in the interiors of Uranus and Neptune. Moreover, the water-rich compound $MgO_4H_6$ could be stable inside the early Earth and therefore may serve as a possible early Earth water reservoir. Our findings, in the poorly explored megabar pressure regime, provide constraints for interior and evolution models of wet planets in our solar system and beyond.

Understanding the reaction between water and rock-forming minerals is of fundamental importance in planetary astrophysics, since rocks and ices are abundant in the interiors of planets, in our solar system, and extrasolar planets[1–7]. The distribution of elements within planetary interiors can have several effects on the observed properties of planets. Hidden elements, usually gases or volatiles, that are locked in refractories in the deep interiors have implications on the short- and long-term evolution of planets, and therefore on their observed properties of radius–mass relation and atmospheric composition[8–11]. In our solar system Uranus and Neptune, the so-called "ice giants", may

contain significant amounts of icy materials, by virtue of their distance from the Sun[7]. Mysteries around the ice giants, such as the non-dipolar and non-axisymmetric magnetic fields[12] and the low luminosity of Uranus[13], highlight the need for more information about the composition and its properties in the interior conditions.

Water is expected to be abundant in exoplanets that were formed beyond the water ice-line[14]. The processes of migration of planets inwards in the protoplanetary disk phase suggest that some of the observed close-in planets contain a substantial amount of water[15]. Thus, the water–rock interaction and its properties are not only

[1]National Laboratory of Solid State Microstructures, School of Physics and Collaborative Innovation Center of Advanced Microstructures, Nanjing University, 210093 Nanjing, China. [2]Astrophysics Research Center of the Open University (ARCO), The Open University of Israel, 4353701 Raanana, Israel. [3]Theory of Condensed Matter Group, Cavendish Laboratory, J. J. Thomson Avenue, Cambridge CB3 0HE, UK. [4]Advanced Institute for Materials Research, Tohoku University 2-1-1 Katahira, Aoba, Sendai 980-8577, Japan. [5]These authors contributed equally: Shuning Pan, Tianheng Huang. ✉e-mail: jiansun@nju.edu.cn

applicable to Uranus and Neptune, but also to the hundreds of intermediate-mass exoplanets that may have water-rich interiors. A classical model for water-rich intermediate-mass planets, like Uranus and Neptune, is usually structured in three layers[3]: a gas envelope consisting of hydrogen and helium; an ice (volatiles, mostly water) layer beneath it, and a rocky core of heavier (refractory) elements in the center. Newer models suggest interior structures with a gradual compositional distribution, based on formation models[16–19] in agreement with Uranus and Neptune measurements[20,21]. However, all the models lack sufficient knowledge of composition interaction at high pressure. In practice, it is still unclear whether the transitions between different elements are sharp or continuous[7,22,23]. The recent prediction of He-H$_2$O compounds[24], H$_3$O[25], and SiO$_2$–H$_2$O compounds[26] indicate that the composition of these layers is much more complex than in the simple layered models. Therefore, improved planetary interior models require better knowledge of chemical interactions between interior elements, which is a key for long-term composition distribution in interiors.

The study of conductivity of water-rock compounds is an important parameter for the thermal evolution of the solar system ice-giants, as well as for wet exoplanets. In terrestrial planets the thermal conductivity of the lower mantle determines the rate at which heat flows across the core–mantle boundary, thus consequently influencing the evolution of both the mantle and the core[27]. As an end-member of the (Mg,Fe)O ferropericlase and an effective heat conductor, MgO has been widely considered as a model mineral for estimating the thermal conductivity of the Earth's mantle[28]. However, the thermal transport properties of the MgO–H$_2$O compounds at extreme conditions have not been investigated yet. To the best of our knowledge, this is the first simulation report on the thermal conductivities of the MgO–H$_2$O compounds in planetary interior conditions.

Further, it has been reported that the amount of water in the Earth's mantle is at least two times the mass of the ocean[29], as a consequence of the high water storage capacity of minerals[30,31]. However, according to models of Solar System the existence of so much water on Earth is abnormal[32], as water is in a vapor form in the inner solar system. Many models are proposed to explain the abundance of water in the Earth, among them Enstatite chondrites[32] and carbonaceous chondrites[33] as the carrier of water into the young Earth. Measurement of the D/H ratio in lava samples revealed that the water in the Earth's deep mantle may originate from protosolar nebula, being stored in the rock[34]. Understanding the origin, transportation, and storage of water in the deep Earth requires the study of hydrous minerals, such as brucite (Mg(OH)$_2$), serpentine (Mg$_3$Si$_2$O$_9$H$_4$), chondrodite (Mg$_5$Si$_2$O$_{10}$H$_2$) and phase B (Mg$_{12}$Si$_4$O$_{21}$H$_2$). It was suggested that δ-AlOOH[35] and Mg(OH)$_2$[36] within the subduction slab can deliver water into the Earth's interior. A recent study[37] finds two olivine-water compounds at the Earth's core condition, which could host much of Earth's water in the first 50–100 million years of its history.

MgO is one of the most abundant rocks in the interiors of planets[38–44]. Interestingly, MgO exhibits high solubility in water (200–400 g/l) between 24–38 GPa, which decreases after 55 GPa[45]. At ambient conditions, MgO and H$_2$O combine to form brucite (Mg(OH)$_2$), which has a layered structure with space group symmetry $P\bar{3}m$1. Upon compression, the protons become disordered[46] and its symmetry reduces to $P\bar{3}$[47]. At 18 GPa, brucite transforms to a $P4_12_12$

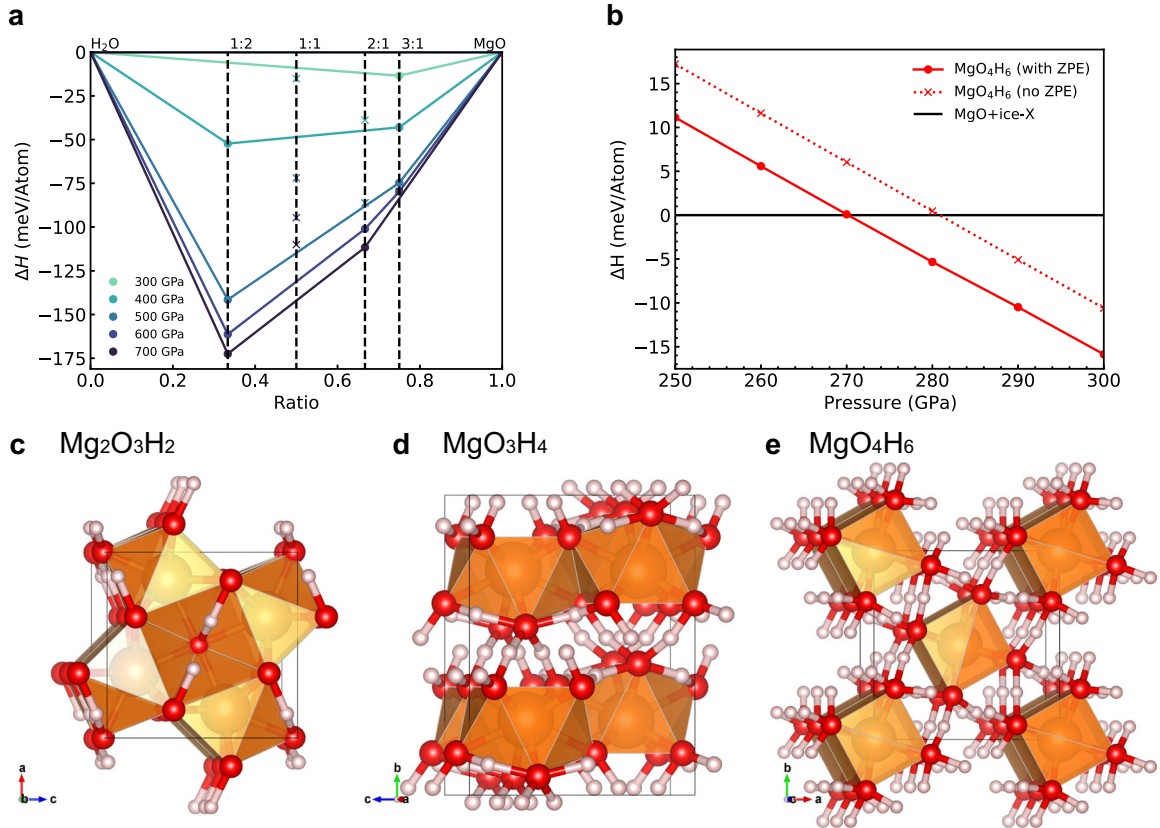

**Fig. 1 | Formation enthalpy and crystal structure of MgO–H$_2$O compounds.**
**a** Convex hull of formation enthalpy at 300–900 GPa. The circle symbols represent stable compositions. The x symbols represent the compositions that decompose. The zero-point energy of phonons is considered in this figure. For MgO, we consider its B1 and B2 phases. For water, we consider all the stable phases mentioned in Ref. [74] including ice X, *Pbcm*, *Pbca*, and *P*3$_1$21 phase. **b** The formation enthalpy of MgO$_4$H$_6$ at 250–300 GPa. With ZPE, MgO$_4$H$_6$ becomes stable at 270 GPa. **c**–**e** The crystal structure of Mg$_2$O$_3$H$_2$, MgO$_3$H$_4$, and MgO$_4$H$_6$, respectively. The orange, red and white spheres correspond to Mg, O, and H atoms.

phase[36] with a three-dimensional network structure. When pressure reaches 27 GPa, brucite decomposes into MgO and ice-VII[36]. It is yet unclear whether there are other stable MgO–$H_2O$ compounds at a higher pressure of hundreds of GPa in planetary interiors.

In this study, with crystal structure predictions, we find three new MgO–$H_2O$ compounds at ultrahigh pressure: $Mg_2O_3H_2$ above 400 GPa, $MgO_3H_4$ above 600 GPa, and $MgO_4H_6$ in the pressure range of 270–600 GPa. Ab initio and machine-learning molecular dynamics simulations reveal their superionic behavior at the pressure-temperature conditions as in the interiors of Uranus and Neptune. Furthermore, we calculate their thermal conductivity and electronic conductivity, which may have effects on the magnetic field and luminosity of Uranus and Neptune.

## Results
### Prediction of new structures
Here, we perform variable-composition structure predictions on MgO–$H_2O$ binary system at 200, 300, 500, and 1000 GPa. Our structure prediction calculations show three stable MgO–$H_2O$ compounds: $Mg_2O_3H_2$, $MgO_3H_4$, and $MgO_4H_6$ (Fig. 1c–e). According to the convex hull of normalized formation enthalpies, these compounds are stable at above 400 GPa, above 600 GPa, and between 270 and 600 GPa, respectively (Fig. 1a, b). The detailed structure parameters are provided in the supplementary material. We also confirmed the convex hull using other calculation methods, as shown in Fig. S2. The inclusion of zero-point energy only slightly changes the stable pressure of $MgO_4H_6$ from 270 GPa to about 280 GPa (Fig. 1b), indicating that the influence from the nuclei quantum effect on the stability of these compounds is small. Phonon calculations show that all three phases are dynamically stable, exhibiting no imaginary phonon modes. Electronic structure calculations show that all three phases are insulators with large band gaps, as shown in Fig. S3.

In all three phases, H atoms form two symmetrized bonds with O atoms. The symmetrization of H bonds is common in high-pressure ice phases, such as ice-VII[48] and ice-X[49]. Note that the Mg atoms in $MgO_3H_4$ are nine-fold coordinated by O atoms, which is a rare phenomenon. The coordination numbers of MgO-B1 and MgO-B2 phases are six and eight, respectively. According to Li et al.[37], the only nine-fold coordinated Mg found in minerals is in $\beta$-$Mg_2SiO_5H_2$. The high coordination of $MgO_3H_4$ confirms their hypothesis[37] that hydration can increase the coordination of Mg.

The water-rich phase $MgO_4H_6$ predicted in this work has an extremely high water storage capacity. It contains 57.3 wt % of water, which is far higher than any reported hydrous minerals, including recently found 11.4 wt % in $Mg_2SiO_5H_2$[37], ~15 wt % in $\delta$-$AlO_2H$ and $MgSiO_4H_2$. In Fig. S4 we show the pressure–density relationship. The densities of all three phases are between those of MgO and $H_2O$. As expected, their densities increase with their MgO ratio. A comparison of the density of each compound to an equivalent MgO–water mixture according to the Additive Volume Law (AVL) shows differences of up to 4% in density, where AVL is usually higher than the reported density.

### Superionic behavior in MgO–$H_2O$ compounds
We carried out ab initio molecular dynamics to explore the state of these phases under high temperature, corresponding to the interior of super-Earth to Neptune mass planets. We follow the mean square displacement (MSD) of the MD trajectories (Fig. 2a–c). All three phases exhibit similar behavior: at low temperature, the MSD slopes for all the atoms are zero, which means that they stay near their equilibrium positions and the sample is in the solid state. On increasing the temperature, the slope of MSDs of the Mg and O atoms remains zero, while the MSD slope of the H atoms becomes none-zero, which means that H atoms leave their equilibrium positions and diffuse like a liquid in the sublattice form by Mg and O atoms. The snapshots of the MD trajec-

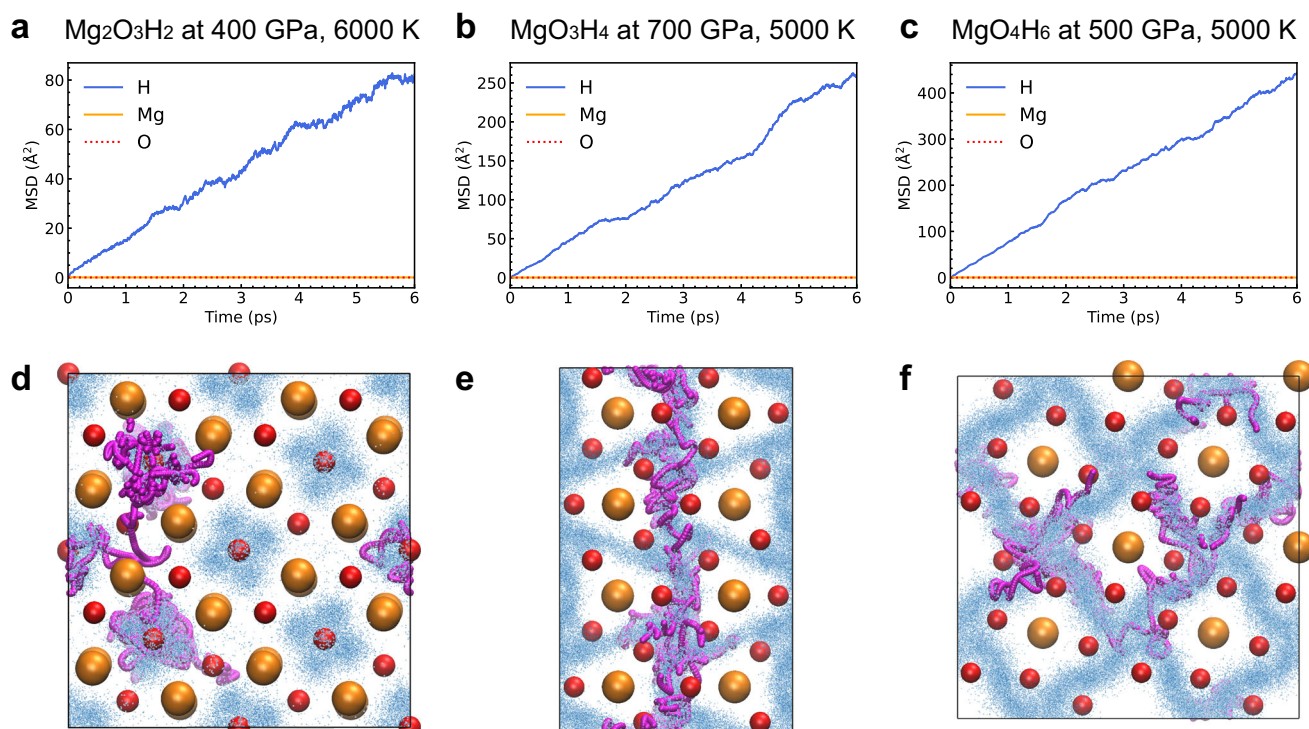

**Fig. 2 | The dynamic properties of MgO–$H_2O$ compounds from molecular dynamics simulations. a** The mean square displacement (MSD) of $Mg_2O_3H_2$ at 400 GPa, 6000 K (the isentropic condition of Uranus). **b** The MSD of $MgO_3H_4$ at 700 GPa, 5000 K (the isentropic condition of Neptune). **c** The MSD of $MgO_4H_6$ at 500 GPa, 5000 K (the isentropic condition of Uranus). **d–f** The snapshots of MD trajectories. The orange and red spheres correspond to Mg and O atoms. The blue points represent H atoms. The position of H atoms from 1000 frames is shown in the picture. We use magenta spheres to highlight the trajectory of a single H atom.

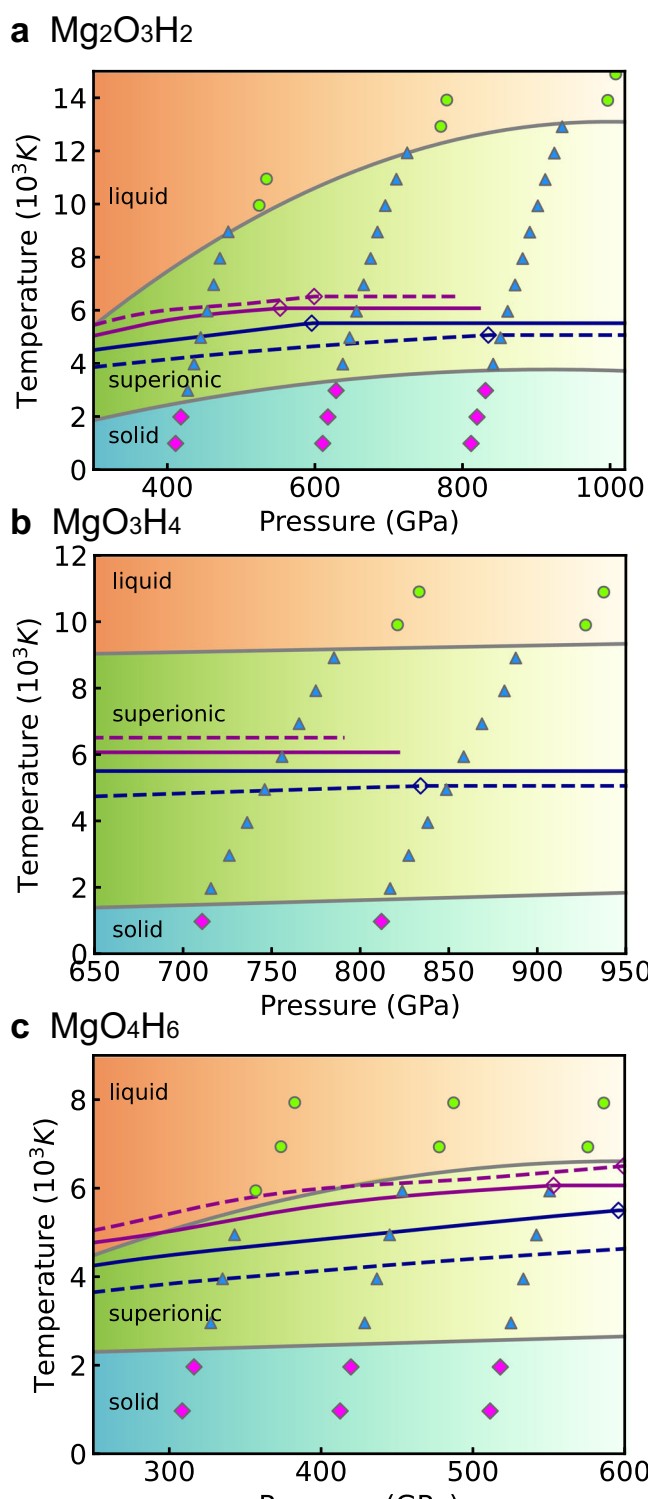

**a** Mg₂O₃H₂

**b** MgO₃H₄

**c** MgO₄H₆

**Fig. 3 | The phase diagrams of MgO–H₂O compounds. a** Mg$_2$O$_3$H$_2$, **b** MgO$_3$H$_4$ and **c** MgO$_4$H$_6$. The purple and blue lines represent the isentropic lines of Uranus and Neptune. The solid and dashed purple line corresponds to U1 and U2 models. The solid and dashed blue line corresponds to N1 and N2b models. The data of models are from ref. [3]. The diamonds represent the condition of the core–mantle boundary.

lowest in Mg$_2$O$_3$H$_2$, indicating that the diffusion rate increases with H$_2$O content. At higher temperatures, the MSD of all the atoms becomes nonzero, and the sample becomes liquid. The radius distribution function (RDF) and vibrational density of state (VDOS) are shown in Figs. S5–6.

The phase diagrams of the three compounds are shown in Fig. 3a–c, together with the isentropic interior profiles of Uranus and Neptune from Nettelmann et al.[3] The phase diagrams show that Mg$_2$O$_3$H$_2$ and MgO$_4$H$_6$ are superionic in interiors of Uranus and Neptune, while MgO$_3$H$_4$ is not superionic in one (N2b) Neptune model[3]. Non-isentropic models of Uranus and Neptune have much hotter deep interiors and slightly colder outer envelopes, due to slower cooling[51,52]. The deep interiors in non-isentropic models lay above the superionic phase, in the liquid phase. In these models, the stably stratified region below the convective envelope accounts for the magnetic field generator[12].

Mg$_2$O$_3$H$_2$, MgO$_3$H$_4$ and MgO$_4$H$_6$ have distinct superionic temperature and melting point. These properties are closely related to the geometry of crystalline cage formed by Mg–O polyhedral, as shown in Fig. 1c–e. In MgO$_4$H$_6$, Mg-O polyhedral forms one-dimensional chains. Different chains are interconnected by O–H–O bonds. In MgO$_3$H$_4$, Mg-O polyhedral forms two-dimensional layer structure. O–H–O bonds connect different layers. In contrast, the Mg–O polyhedral in Mg$_2$O$_3$H$_2$ connects to all its neighbors by Mg–O bonds. In superionic phase, the O–H bonds can be broken, affecting the stability of chain/layer structures at high temperature. But Mg–O bonds stay intact in superionic phase. Thus, Mg$_2$O$_3$H$_2$ has higher melting point than the other two compounds.

In Fig. 4a–c, we project the mean square displacement (MSD) of protons into the $x$, $y$, and $z$ direction. All three compounds show some degree of anisotropy. In MgO$_3$H$_4$, the preferable diffusion directions are $a$ and $b$ direction, because it is more difficult for protons to move across Mg–O layers. The preferable diffusion direction in MgO$_4$H$_6$ is the $c$ direction, since the diffusion perpendicular to Mg–O chains is hindered. The preferable diffusion direction in Mg$_2$O$_3$H$_2$ is the $b$ direction. However, even in the $b$ direction the diffusion path is blocked by Mg–O polyhedral, which indicates a large energy barrier. This is why Mg$_2$O$_3$H$_2$ has higher solid-superionic temperature and low diffusion rate, compared with MgO$_3$H$_4$ and MgO$_4$H$_6$.

The solid-superionic temperature can also be affected by the degree of localization of protons. In all three compounds, each proton connects to two O atoms, forming (possibly nonsymmetric) O–H–O bonds. We use the three-body analysis in reference[53] to analyze the O–H–O bonds. We find all O–H–O triplets by first find the closest oxygen (O$_a$) and second oxygen (O$_b$) of each H atom. At zero temperature, the difference of two H-O bonds in length (HO$_b$–HO$_a$) is 0.07 Å in Mg$_2$O$_3$H$_2$ (800 GPa), 0.07–0.18 Å in MgO$_3$H$_4$ (800 GPa), 0–0.02 Å in MgO$_4$H$_6$ (600 GPa). Among them, the O–H–O bonds are the most nonsymmetric in MgO$_3$H$_4$, and most symmetric in MgO$_4$H$_6$. At finite temperature, the spatial distribution of protons in Fig. 4d–f shows that the protons in MgO$_3$H$_4$ have higher degree of delocalization compared with MgO$_4$H$_6$. Thus, the protons in MgO$_3$H$_4$ are less confined to a O–H–O triplet, making solid-superionic temperature lower.

The proton flow of the superionic phase can generate electrical transport. We employed the Nernst-Einstein equation to calculate the ionic electrical conductivity of MgO–H$_2$O compounds under the core–mantle boundary condition of Uranus and Neptune. As is shown in Fig. 5b, in the temperature range from 1000 K to 6000 K at 600 GPa, the electrical conductivity of Mg$_2$O$_3$H$_2$ is 0.36–11.07/Ω cm in Mg$_2$O$_3$H$_2$, 26–93/Ω cm for MgO$_3$H$_4$, and 26–137/Ω cm for MgO$_4$H$_6$. The electrical conductivity of Mg$_2$O$_3$H$_2$ is much lower than H$_2$O (120–350/Ω cm)[2]. In MgO$_3$H$_4$ and MgO$_4$H$_6$, the electrical conductivities are also lower than H$_2$O, but they are in the same order of magnitude. More details concerning these calculations are provided in the supplementary material.

tories (Fig. 2d–f) show that the spatial distribution of different H atoms overlaps. Such a state is known as the 'superionic state'[50], which is common in hydrates under high pressure. More discussion about the definition of superionic state is available in the supplementary information. The proton diffusion rate is the highest in MgO$_4$H$_6$ and the

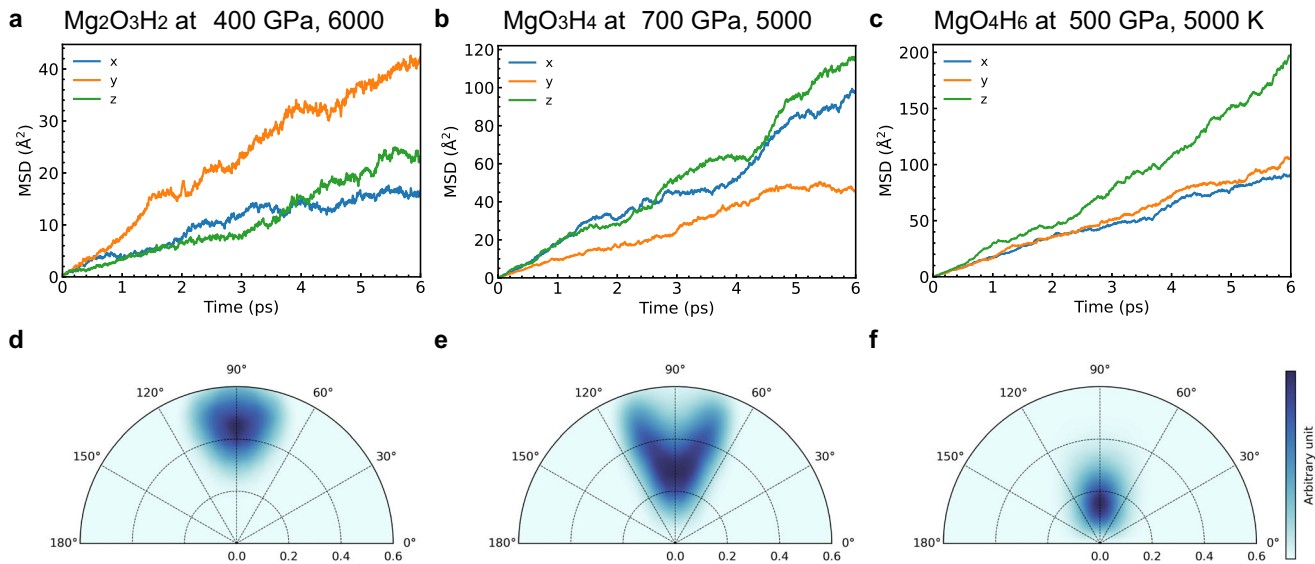

**Fig. 4 | Analysis for superionic behavior in MgO–H₂O compounds. a** The projected mean square displacement (MSD) of $Mg_2O_3H_2$ at 400 GPa, 6000 K. **b** The projected MSD of $MgO_3H_4$ at 700 GPa, 5000 K. **c** The projected MSD of $MgO_4H_6$ at 500 GPa, 5000 K. **d**–**f** Spatial distribution of protons in $Mg_2O_3H_2$, $MgO_3H_4$, and $MgO_4H_6$ at 1000 K. We use M to denote the midpoint of $O_aO_b$. The radial distance represents the length of HM in Å. The angle represents the angle formed by $O_aO_b$ and HM.

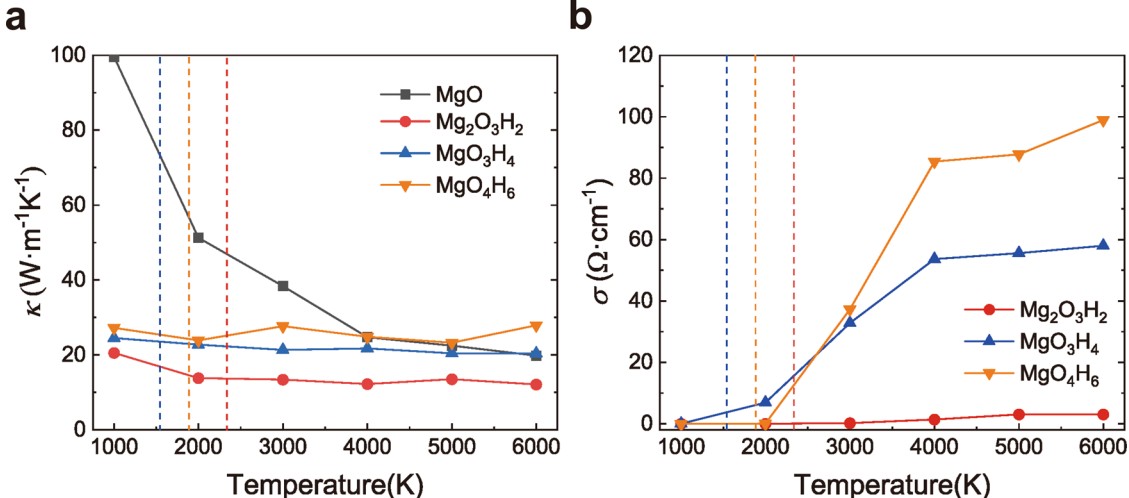

**Fig. 5 | The transport properties of the MgO–H₂O compounds (a-b). a**, **b** Green, red, blue, and black lines represent the $MgO_4H_6$, the $MgO_3H_4$, the $Mg_2O_3H_2$, and the MgO at 600 GPa, respectively. The green, red, and blue dashed lines represent the phase boundary of the $MgO_4H_6$, $MgO_3H_4$, and $Mg_2O_3H_2$, respectively. $\kappa$ means heat conductivity and $\sigma$ means electronic conductivity.

## Thermal conductivity

The distribution of the three compounds in the interior depends on their density. Since stable $MgO_4H_6$ has a high water ratio and thus lower density than the other two compounds, as is shown in Fig. S2, it is located near the pure ice layer. On the contrary, $Mg_2O_3H_2$ is stable above 400 GPa pressure and has a lower water ratio, and thus is more favorable in the deeper region near the rocky core. The compounds we found indicate that rocky core erosion is supported by the ice-rock interaction with the surrounding ice layer, forming a composition gradient. In Fig. 6 we present such possible structures for Uranus and Neptune, based on the models of Nettelmann et al.[3]

Composition distribution has implications for heat transport. The composition gradients shown in Fig. 6 are of decreasing outwards mean molecular weight, which may suppress heat transport by large-scale convection[54,55]. The low luminosity (surface heat flux) of Uranus means that its surface is very cold. So, either all its heat has

been lost, or the heat is captured inside. It has been hypothesized that some form of thermal boundary slows down the cooling process. One explanation for the low luminosity of Uranus is a composition gradient between the metal-rich (ice/rock) interior to the gas envelope, which suppresses convection, hence the heat is trapped in the deep interior while the surface is cold[51,52]. Here we suggest a compositional gradient between a rock-rich deep interior and an ice-rich upper layer. Although this scenario requires some further investigation, under certain conditions it may perform as a barrier for heat transport to explain Uranus' low luminosity. The exact composition gradient and its effect on heat transport is beyond the scope of this paper, more details can be found in dedicated works[45,51,52,54–56].

When convection is suppressed by a composition gradient in the interior, the heat transport is via layered convection[51] and/or conduction. We thus investigated the thermal conductivities of the MgO–H₂O

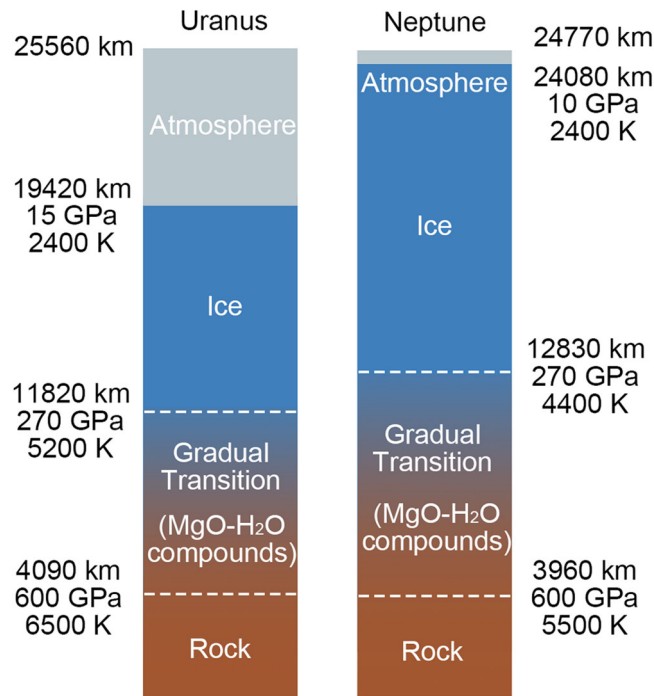

**Fig. 6 | Suggested interior models for internal structure models of Uranus and Neptune based on the data from the U2 and N1 models in Ref. [3].** The blue-brown gradual change region is where MgO–$H_2O$ compounds can exist.

compounds in the temperature range from 1000 K to 6000 K and compared them with the thermal conductivities of the B2 phase MgO. As shown in Fig. 5a, the thermal conductivities of $Mg_2O_3H_2$, $MgO_3H_4$ and $MgO_4H_6$ at 600 GPa and 3000 K are 13.439, 21.323, and 23.590 W $m^{-1}$ $K^{-1}$, respectively, positively related to the water content. These values are evidently lower than the thermal conductivities of the MgO under the same conditions. In addition, all three compounds have a distinct property that their thermal conductivities do not change much with increasing temperature in the range from 3000 K to 6000 K, in which these compounds remain in a superionic state. The states of the MgO–$H_2O$ compounds significantly affect their thermal conductivities. Figure 5a shows that in the range from 1000 K to 2000 K the thermal conductivities of all three compounds decrease with the increasing temperature, which is consistent with other solids but different from their behavior in the superionic state.

## Discussion

The abundance of different compositions in planets are still an open question. The ice to rock ratio in interior models of Uranus and Neptune is degenerated, varying from rock dominated to ice dominated models[57]. Based on planet formation models, abundances of rocks and ices are expected to be similar in mass exterior to the ice-line (water condensation line). Interior models of planets suggest that rocky composition mainly consist of Mg, Si, and Fe compounds[38], where MgO and $SiO_2$ are the most basic compounds. From the abundance of chemical elements in the universe[58], we can see that the abundance of Mg, Si and Fe are very similar. MgO is also the main topic in some recent studies[22,45] about planet models. As a result, we can safely assume that MgO as one of the main components of the rocky core. $H_2O$ is the most abundant ice, and the first to condense in the proto-planetary disk, thus it is expected to be main components of "hot ice" layer.

The superionic behavior of rock compounds with water was reported in several systems and isn't unique to MgO rock. Silica and water may form superionic $Si_2O_5H_2$ at conditions as of the core–mantle

boundary of Uranus and Neptune[24]. Under a hydrogen-rich environment, water forms $H_3O$ and still has superionic behavior[25]. As recent experimental work[45] showed that MgO is highly soluble in water at conditions such as in the upper layers of Uranus and Neptune, the MgO–$H_2O$ ionization phases at higher pressure have to be explored. In this research, we find that all three stable MgO–$H_2O$ compounds exhibit superionic behavior at the condition in interiors of super-Earths and Neptunes. We infer that the superionic behavior of water is consistent with compositional inhomogeneity, such as the enrichment of $SiO_2$, MgO, and H.

In Fig. 6 we sketched the internal structure models of Uranus and Neptune based on the predicted MgO–$H_2O$ compounds, on top of the data from the U2 and N1 model in Nettelmann et al.[3]. In traditional three-layer model[3], there is a boundary between rock core and ice layer and the density profile is discontinuous at the boundary. It was shown that a gradual transition model should be able to explain the observational results[59]. Our prediction of MgO–$H_2O$ compounds indicates that the ice layer can erode the MgO rock in the core and results in a "fuzzier" boundary. This is in favor of the gradual transition model. However, the structures shown in Fig. 6 are suggestions, and the exact composition distribution in Uranus and Neptune are beyond the scope of this paper and requires further investigation.

The superionic state of the MgO–$H_2O$ compounds has lower electrical conductivity in comparison to pure water at the same conditions. However, the compounds found in this work are stable at higher pressure, i.e., much deeper in the interior than the predicted location of the magnetic field generator in Uranus and Neptune[12]. In this case, as is emphasized in Fig. 5, the magnetic field is generated in an upper pure water layer.

In addition to playing a central role in the evolution of the ice giants, the MgO–$H_2O$ compounds found here are relevant also for water-rich exoplanets. The deep interiors of exoplanets in the mass range of super-Earths to Neptunes are found to be similar for further out planets and their planetary twins that have migrated inwards[23]. Therefore, the compound found here are relevant also for migrating in water-rich exoplanets. Interior models of exoplanets with mixed ice and rock were recently suggested[11,23], but lack of knowledge on ice-rock interaction at 100 s GPa pressure. The three compounds found here indicate that high water content, of up to 57.3 wt%, is possible in the deep interiors of wet planets in the super-Earth to Neptune mass range. Such planets may have surface oceans or steam atmospheres caused by the ice-rock separation at low pressures[23], and fit the recent discovery of exoplanets around M dwarfs with densities of 50% rock and 50% water[15].

Furthermore, the water-rich compound $MgO_4H_6$ may help explain the origin of water on the Earth. Magnesium hydrosilicate compounds are found to be able to store a large amount of water (up to 8 times the mass of the ocean) in the Earth's depths[37]. Similarly, the $MgO_4H_6$ we find in this study is stable above 270 GPa, suggesting that water could be stored in the central region of the early Earth in the form of $MgO_4H_6$. According to this scenario, the dense iron alloys that sunk to the Earth's core moved up the $MgO_4H_6$ to a shallower region, where it decomposed and released the water: $MgO_4H_6 \rightarrow MgO + 3H_2O$.

In summary, we find three new sable MgO–$H_2O$ compounds at megabar pressures. One of the compounds has the highest reported water content of 57.3 wt%. All three phases exhibit superionic behavior under the P-T conditions corresponding to the interior of planets and are consistent with the properties of Uranus and Neptune. The $MgO_4H_6$ compound may also serve as one of the Earth's early water reservoirs. We conclude that MgO-water mixtures are likely in deep interiors of water-rich super-Earth to Neptune mass planets, where a large fraction of the water can be locked in MgO rocks in the deep interior. The existence of various compounds supports gradual composition distribution in planetary interiors.

## Methods

### Structure prediction

The structures of the $(MgO)_x$-$(H_2O)_y$ ($x = 1–4$, $y = 1–4$) system are searched by Magus code[60,61] with up to 40 atoms per unit cell. Each generation contains 30 structures, and we run over 60 generations until the result converges. The 40% outcome structures with the lowest enthalpy were used as the seeds for the evolution of the next generation and the left structures in each generation were randomly produced. We also performed some other more complex variable-composition structure predictions on the Mg–O–H ternary system to check the results. The results were cross-checked using the AIRSS[62,63] code.

### Ab initio calculations

The ab initio calculations are performed by the VASP code[64]. We use the standard projector augmented-wave (PAW) method[65] and the Perdew-Burke-Ernzerhof exchange-correlation functional[66] in the generalized gradient approximation. We treat $2s^2 2p^6 3s^2$, $2s^2 2p^4$, and $1s^1$ electrons as valence elections for Mg, O and H atoms. We used a plane-wave energy cutoff of 1050 eV and the k-point sample resolution of $2\pi \times 0.025\ \text{Å}^{-1}$ for the ab initio calculations. The result of the convergence test is provided in the supplementary material (Fig. S1). The results were cross-checked using CASTEP[67,68] code with similar parameters. The phonon and zero-point energy (ZPE) calculations were performed with the PHONOPY code[69].

### Molecular dynamics (MD) simulations

The MD simulation was performed with an NVT ensemble using the Nosé–Hoover thermostat[70] implemented in VASP. For AIMD simulation, we used gamma point for k-point sampling, 600 eV for plane-wave energy cutoff, and standard pseudopotentials. The timestep was set to 0.5 fs to treat the fast-moving protons. The total simulation time was 10 ps. As for the MD with machine-learning neural network potential, we used GPUMD[71] with periodic boundary conditions and a time step of 0.25 fs. The total time of a simulation was at least 1 ns.

### Electrical conductivity

We use Nernst–Einstein equation ($\sigma = DNq^2/k_B T$) to calculate electrical conductivity. In the equation, $q$ is the carrier electric charge ($1e$ for H atoms), $D$ is the carrier diffusion coefficient, $N$ is the carrier density, and $T$ is the temperature. All these values can be extracted from the MD trajectories.

### Thermal conductivity

We used the Green–Kubo formula together with Onsager's phenomenological approach to calculate thermal conductivity, as implemented in Sportrans[72,73]. In this approach the interactions among different conserved fluxes are explicitly accounted for by Onsager's phenomenological relations:

$$J_i = \sum_j \Lambda_{ij} f_j \tag{1}$$

$J$ is a generic conversed flux and $f$ is a thermodynamic affinity. $\Lambda_{ij}$ coefficients are expressed as integrals of the relevant fluxes:

$$\Lambda_{ij} = \frac{\Omega}{k_B} \int_0^\infty \left\langle \mathscr{I}_i(t) \mathscr{I}_j(0) \right\rangle \mathrm{d}t \tag{2}$$

$\mathscr{I}_i(t)$ is the time series of the $i$th flux and can be obtained from the MD simulation. $\Omega$ is the volume of the system and $k_B$ is the Boltzmann constant. Theoretically, $i$th flux represent all the flux which should be considered. In this study, since all three MgO–$H_2O$ compounds are insulator, we did not take electronic thermal flux into consideration.

Another reason for us to ignore the electronic thermal flux is that we used more than 6000 atoms to run the MD simulations, making it impossible to get electron wave functions. So, in this paper $i$th flux represent ionic thermal flux and the lattice thermal flux.

### Machine-Learning neural network potential

We trained a machine-learning neural network potential for the Mg–O–H system using the NEP and GPUMD[71] method, which directly uses relative atomic coordinates to describe local atomic environments to obtain atomic energies, while forces are obtained by taking the derivatives of the energies. Our dataset was built by choosing 16,000 configures in the trajectories from AIMD in the NVT ensemble in the pressure range from 300 GPa to 800 GPa with a temperature step of 1000 K from 3000 to 7000 K.

## Data availability

All data that support the conclusions of this study are included in the article and the Supplementary Information file. These data are available from the corresponding author upon request.

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

## Acknowledgements

J.S. gratefully acknowledges the financial support from the National Key R&D Program of China (grant nos. 2022YFA1403201), the National Natural Science Foundation of China (grant nos. 12125404, 11974162, and 11834006), and the Fundamental Research Funds for the Central Universities. C.J.P is supported by the EPSRC through grants EP/P022596/1, and EP/S021981/1; A.V. acknowledges support by the ISF through grants 770/21 and 773/21. The calculations were carried out using supercomputers at the High Performance Computing Center of Collaborative Innovation Center of Advanced Microstructures, the high-performance supercomputing center of Nanjing University.

## Author contributions

J.S. conceived and led the project. S.P., T.H., Z.L., J.W., and C.J.P. performed the calculations. S.P., T.H., J.S., A.V., C.L., and C.J.P. analyzed the data. S.P., T.H., A.V., J.S., C.J.P., H.-T.W., and D.X. wrote the manuscript. All authors discussed the results and commented on the manuscript.

## Competing interests

C.J.P. is an author of the CASTEP code, and receives royalty payments from its commercial sales by Dassault Systèmes. The remaining authors declare no competing interests.
