## [Peer Review File · Nature Communications]

Magnesium Oxide-Water Compounds at Megabar Pressure and Implications on Planetary InteriorsREVIEWER COMMENTS

Reviewer #1 (Remarks to the Author):

Previous study considered MgO and H₂O are separated at tens of GPa. This paper challenged this concept, proposed this system to hundreds of GPa and found several meaningful compounds of Mg₂O₃H₂, MgO₄H₆ and MgO₃H₄. Their results are rather important to the model of ice giants, such as Uranus and Neptune. In this way, I suggest it to be published in Nat. Commun.

But their papers could be improved, and my comments are as following:

Phase diagram

1. Compared with Mg₂O₃H₂ and MgO₄H₆, MgO₃H₄ seems to have a lower solid-superionic temperature and higher superionic-liquid temperature. It is an interesting phenomenon. But why?

Electronic conductivity,

2. Can the authors explain why H₂O has a much higher proton conductivity than MgO-H₂O compounds?

Thermal conductivity

3. More discussion is needed for relationship between low luminosity. Why would the planet have low luminosity if it has a heat barrier inside? Are there any related references?

4. Why MgO-H₂O has lower thermal conductivity than MgO could be discussed in physics with more details.

5. Their description in method to get thermal conductivity is confused. What does i-th flux represent? Is it for ionic thermal flux or electronic thermal flux?

For the planet model,

6. Why the authors believed there is a gradual transition zone between Ice and MgO-H₂O layers?

Based their calculation, there should be a boundary not a gradual transition zone.

For Figures

7. What κ and σ represent should be given in the title of Fig 4

Reviewer #2 (Remarks to the Author):

In this paper, the authors combine the crystal structure prediction method with MD simulation techniques to study a set of new Mg-O-H compounds that will be thermodynamically stable at the pressure condition close to the interiors of Earth, Uranus and Neptune. They propose that the existence of such water-bearing compounds can be used to refine the evolution models of wet planets. The authors are very experienced in this area. Similar computational techniques have been applied to many other systems by the same group of authors in the recent years. Therefore, I have no technical comments on this manuscript. It can be published without major revision if the editor believe that the topic fit the scope of Nat. Comm.

That said, I have some reservation about the readership of this manuscript. The results look fancy to the general readers when people just started to employ crystal structure prediction to study the planetary minerals. However, the authors have published tens of similar works in general physics journals like Nat. Phys., Phys. Rev. series and PNAS in the past decade. To make a true impact, the authors should submit their work to more specialized journals like Astrophysics, Icarus, J. Geophys. Res., Mon. Not. R. Astron. Soc., or Nat. Astronomy.

Reviewer #3 (Remarks to the Author):

The authors report a prediction of three new magnesium oxide-water compounds at megabar pressures. Magnesium and water do react to form Mg(OH)₂ at condition, but the product decomposes at 27 GPa. The authors found, using multiple structure prediction methods, that magnesium and water can react again at ultrahigh pressures, specifically, Mg₂O₃H₂ would form above 400 GPa, MgO₃H₄

above 700GPa, and MgO₄H₆ in the pressure range of 270-600 GPa. These conditions are close to the pressure conditions of the interior of several planets. Therefore, this finding suggests that water may be stored in MgO rock in the deep interiors of Earth to Neptune mass planets. I think this is a very interesting idea that is suitable for publication in Nature Communications. In particular, one of the predicted compounds, MgO₄H₆, is estimated to contain 57.3 wt % of water, much higher than the previously reported hydrous minerals. I have a few comments that the authors may consider.

(1) The authors features 'superionic' behaviors of the MgO-H₂O compounds prominently. I am wondering whether there is a quantifiable definition of superionicity? The molecular dynamics map does show the protons are all over the place, but I do not believe we should eyeball a superionic state.

(2) What is the abundance of MgO and H₂O in the interior of the planets? There must be a reasonable amount of both for the current prediction to be relevant.

(3) The predicted compounds contains a large amount of hydrogen, and therefore quantum effects are expected to be significant. Would the VASP MD and machine learning version of AIMD still be adequate to describe the behaviors of these compounds, in particular the dynamics of hydrogen? This part of simulation is perhaps more appropriate done with other methods such as path integral molecular dynamics. The authors may comment or justify their choice of method.

Manuscript ID: NCOMMS-22-45591

Magnesium Oxide-Water Compounds at Megabar Pressure and Implications on Planetary Interiors

By Shuning Pan *et al.*

Response to Reviewers

Reviewer #1 (Remarks to the Author):

Previous study considered MgO and H₂O are separated at tens of GPa. This paper challenged this concept, proposed this system to hundreds of GPa and found several meaningful compounds of Mg₂O₃H₂, MgO₄H₆ and MgO₃H₄. Their results are rather important to the model of ice giants, such as Uranus and Neptune. In this way, I suggest it to be published in Nat. Commun.

But their papers could be improved, and my comments are as following:

Phase diagram

1. Compared with Mg₂O₃H₂ and MgO₄H₆, MgO₃H₄ seems to have a lower solid-superionic temperature and higher superionic-liquid temperature. It is an interesting phenomenon. But why?

Reply: About the superionic-liquid temperature (melting point): as shown in Figure 3 (Fig.4 in the revised manuscript), the superionic-liquid temperature of Mg₂O₃H₂ is the highest among three MgO-H₂O compounds. At 500 GPa, the melting point of Mg₂O₃H₂ is around 9000 K, which is higher than 6000 K for MgO₄H₆. At 800 GPa, the melting point of Mg₂O₃H₂ is around 13000 K, which is higher than 9000 K for MgO₃H₄. The melting point of MgO₃H₄ and MgO₄H₆ cannot be compared directly because of their distinct stable pressure range (MgO₄H₆ is stable at 270-600 GPa, while MgO₃H₄ is stable above 600 GPa).

The difference in melting point can be attributed to the geometry of Mg-O framework. Figure R1 displays the Mg-O polyhedral in MgO-H₂O compounds. In

MgO₄H₆, Mg-O polyhedral forms one-dimensional chains. Different chains are interconnected by O-H-O bonds. In MgO₃H₄, Mg-O polyhedral forms two-dimensional layer structure. O-H-O bonds connect different layers. In contrast, the Mg-O polyhedral in Mg₂O₃H₂ connects to all its neighbors by Mg-O bonds. In superionic phase, the O-H bonds can be broken, affecting the stability of chain/layer structures at high temperature. But Mg-O bonds stay intact in superionic phase. Thus, Mg₂O₃H₂ has higher melting point than the other two compounds.

Figure R1 | Crystal structure of MgO-H₂O compounds.

About the solid-superionic temperature: the superionic behavior of MgO-H₂O compounds is closely related to the geometry of crystalline cage formed by Mg-O polyhedral. In Figure R2 (a)-(c), we project the mean square displacement (MSD) of protons into three cell vectors (*a*, *b* and *c* axis). All three compounds show some degree of anisotropy. In MgO₃H₄, the preferable diffusion directions are *a* and *b* direction, because it is more difficult for protons to move across Mg-O layers. The preferable diffusion direction in MgO₄H₆ is *c* direction, since the diffusion perpendicular to Mg-O chains is hindered. The preferable diffusion direction in Mg₂O₃H₂ is *b* direction. However, even in *b* direction the diffusion path is blocked by Mg-O polyhedral, which indicates a large energy barrier. This is why Mg₂O₃H₂ has higher solid-superionic temperature and low diffusion rate, compared with MgO₃H₄, and MgO₄H₆.

Figure R2 | Analysis for superionic behavior in MgO-H₂O compounds. (a) The projected MSD of Mg₂O₃H₂ at 400 GPa, 6000 K (b) The projected MSD of MgO₃H₄ at 700 GPa, 5000 K. (c) The projected MSD of MgO₄H₆ at 500 GPa, 5000 K. (d)-(f) Spatial distribution of protons in Mg₂O₃H₂, MgO₃H₄, and MgO₄H₆ at 1000 K. We use M to denote the midpoint of O_aO_b. The radial distance represents the length of HM in Å. The angle represents the angle formed by O_aO_b and HM.

The solid-superionic temperature can also be affected by the degree of localization of protons. In all three compounds, each proton connects to two O atoms, forming (possibly nonsymmetric) O-H-O bonds. We use the three-body analysis in reference [*Phys. Rev. Lett.* **117**, 1–5 (2016)] to analyze the O-H-O bonds. We find all O-H-O triplets by first find the closest oxygen (O_a) and second oxygen (O_b) of each H atom. At zero temperature, the difference of two H-O bonds in length (HO_b-HO_a) is 0.07Å in Mg₂O₃H₂ (800 GPa), 0.07~0.18Å in MgO₃H₄ (800 GPa), 0-0.02 Å in MgO₄H₆ (600 GPa). Among them, the O-H-O bonds are the most nonsymmetric in MgO₃H₄, and most symmetric in MgO₄H₆. At finite temperature, the spatial distribution of protons in Figure R2 (d)-(f) shows that the protons in MgO₃H₄ have higher degree of delocalization compared with MgO₄H₆. Thus, the protons in MgO₃H₄ are less confined to a O-H-O triplet, making solid-superionic temperature lower.

We thank the referee for raising this point, which made us to perform deeper analysis. We have added these discussions into the revised manuscript.

Electronic conductivity,

2. Can the authors explain why H₂O has a much higher proton conductivity than MgO-H₂O compounds?

Reply: We've updated the electronic conductivity data in the manuscript. As is shown in Fig. 5. (b), in the temperature range from 1000 K to 6000 K at 600 GPa, the electrical conductivity of Mg₂O₃H₂ is 0.36-11.07/Ωcm in Mg₂O₃H₂, 26-93/Ωcm for MgO₃H₄, and 26-137/Ωcm for MgO₄H₆. The electrical conductivity of Mg₂O₃H₂ is much lower than H₂O (120-350/Ωcm) [*Icarus* **211**, 798–803 (2011)]. In MgO₃H₄ and MgO₄H₆, the electrical conductivities are also lower than H₂O, but they are in same order of magnitude.

We calculated protonic conductivity by Nernst–Einstein equation ($\sigma = DNq^2/k_B T$). In the equation, q is the carrier electric charge, D is the carrier diffusion coefficient, N is the carrier density, and T is temperature. Superionic phases of H₂O and MgO-H₂O compounds have the same charge carriers (protons), so q has no major effect on their protonic conductivity. The temperature condition of superionic H₂O conductivity data [*Icarus* **211**, 798–803 (2011)] is around 4000~6000 K. Our calculations cover this temperature range, so temperature does not cause the difference.

The carrier densities of MgO-H₂O compounds are lower than H₂O. At 400 GPa, the carrier density is 0.303/Å³ in *Pbcm* phase [*Phys. Rev. Lett.* **110**, 1–5 (2013)] of H₂O, 0.079/Å³ in Mg₂O₃H₂, 0.207/Å³ in MgO₄H₆ (0.178/Å³ in MgO₃H₄, but it is not stable at 400 GPa). Among them, H₂O has the highest carrier density, which is in favor of proton conductivity.

The proton diffusion rates of MgO-H₂O compounds are also lower than H₂O. At 3000 K, the proton diffusion rate is around 7×10⁻⁸ m²/s in H₂O [*Nat. Phys.* **17**, 1228–1232 (2021)], which is about three times the value of MgO₄H₆ and MgO₃H₄, 20 times the value of Mg₂O₃H₂. This can be explained by the existence of Mg-O polyhedral, which hinders the proton diffusion (this is also discussed in the last question). Similar results can be found in MgO-SiO₂-H₂O system [*Phys. Rev. Lett.* **128**, 035703 (2022)]

and SiO₂-H₂O system [*Phys. Rev. Lett.* **128**, 35702 (2022)], where the proton diffusion rates are also lower than in pure H₂O. We've added this discussion to supplemental information.

Thermal conductivity

3. *More discussion is needed for relationship between low luminosity. Why would the planet have low luminosity if it has a heat barrier inside? Are there any related references?*

Reply: Basically, the low luminosity (surface heat flux) of Uranus means that its surface is very cold. So, either all the its heat has been lost, or the heat is captured inside. It has been hypothesized that some form of thermal boundary slows down the cooling process. A gradual composition distribution can serve as such as a heat barrier, by suppressing the heat convection. We think that the relationship between low luminosity and the heat barrier is not the major topic of our paper, so we only make a brief discussion. More details can be found in references papers [*Space Sci. Rev.* **152**, 423–447 (2010); *Astron. Astrophys.* **633**, 1–10 (2020); *Astron. Astrophys.* **650**, 1–12 (2021); *Nat. Astron.* **5**, 815–821 (2021)]. We have added these brief discussions and the references into the revised manuscript.

4. *Why MgO-H₂O has lower thermal conductivity than MgO could be discussed in physics with more details.*

Reply: At 600 GPa, MgO is solid between 3000~6000 K, while MgO-H₂O compounds are superionic. In solid insulator crystals, the thermal conductivity relates to its crystal structure. The main contributor to heat transport is lattice thermal flux. In superionic phase materials, however, the ion diffusion breaks the periodic lattice, and ionic thermal flux plays the major role in heat transport. In MgO-H₂O compounds, the contribution from the ionic thermal flux cannot compensate for the reduction of lattice heat flux, leading to lower thermal conductivity in MgO-H₂O compounds.

5. Their description in method to get thermal conductivity is confused. What does *i*-th flux represent? Is it for ionic thermal flux or electronic thermal flux?

Reply: Theoretically, *i*-th flux represent all the flux which should be considered. In this study, since all three MgO-H₂O compounds are insulator, we did not take electronic thermal flux into consideration. Another reason for us to ignore the electronic thermal flux is that we used more than 6000 atoms to run the MD simulations, making it impossible to get electron wave functions. So, in this paper *i*-th flux represent ionic thermal flux and the lattice thermal flux.

For the planet model,

6. Why the authors believed there is a gradual transition zone between Ice and MgO-H₂O layers? Based their calculation, there should be a boundary not a gradual transition zone.

Reply: In traditional three-layer model [*Planet. Space Sci.* **77**, 143–151 (2013)], there is a boundary between rock core and ice layer and the density profile is discontinuous at the boundary. It was shown that a gradual transition model should be able to explain the observational results [*Astrophys. J.* **726**, 15 (2011)]. Our prediction of MgO-H₂O compounds indicates that the ice layer can erode the MgO rock in the core and results in a “fuzzier” boundary. This is in favor of the gradual transition model. However, the structures shown in Fig. 6 are suggestions, and the exact composition distribution in Uranus and Neptune are beyond the scope of this paper and requires further investigation. We emphasized this point in the text and in the figure caption.

For Figures

7. What κ and σ represent should be given in the title of Fig 4

Reply: The explanation of κ and σ has been added.

We appreciate all the referee’s valuable comments again, which help us to improve our manuscript substantially.

Reviewer #2 (Remarks to the Author):

In this paper, the authors combine the crystal structure prediction method with MD simulation techniques to study a set of new Mg-O-H compounds that will be thermodynamically stable at the pressure condition close to the interiors of Earth, Uranus and Neptune. They propose that the existence of such water-bearing compounds can be used to refine the evolution models of wet planets. The authors are very experienced in this area. Similar computational techniques have been applied to many other systems by the same group of authors in the recent years. Therefore, I have no technical comments on this manuscript. It can be published without major revision if the editor believe that the topic fit the scope of Nat. Comm.

That said, I have some reservation about the readership of this manuscript. The results look fancy to the general readers when people just started to employ crystal structure prediction to study the planetary minerals. However, the authors have published tens of similar works in general physics journals like Nat. Phys., Phys. Rev. series and PNAS in the past decade. To make a true impact, the authors should submit their work to more specialized journals like Astrophysics, Icarus, J. Geophys. Res., Mon. Not. R. Astron. Soc., or Nat. Astronomy.

Reply: We are glad that the referee knows some of our previous work. And we thank the referee for the overall positive comment on this work. We think this work and in particular the prediction of the most water-rich compound found so far, is a significant progress for exoplanet astrophysics and planetary science and will attract many general readers from different fields, such as physics, materials science, astronomy, planetary science, geochemistry, etc. Therefore, we believe *Nature Communications* is a proper arena for this work.

Reviewer #3 (Remarks to the Author):

The authors report a prediction of three new magnesium oxide-water compounds at megabar pressures. Magnesium and water do react to form Mg(OH)₂ at condition, but the product decomposes at 27 GPa. The authors found, using multiple structure prediction methods, that magnesium and water can react again at ultrahigh pressures, specifically, Mg₂O₃H₂ would form above 400 GPa, MgO₃H₄ above 700GPa, and MgO₄H₆ in the pressure range of 270-600 GPa. These conditions are close to the pressure conditions of the interior of several planets. Therefore, this finding suggests that water may be stored in MgO rock in the deep interiors of Earth to Neptune mass planets. I think this is a very interesting idea that is suitable for publication in Nature Communications. In particular, one of the predicted compounds, MgO₄H₆, is estimated to contain 57.3 wt % of water, much higher than the previously reported hydrous minerals. I have a few comments that the authors may consider.

(1) The authors features ‘superionic’ behaviors of the MgO-H₂O compounds prominently. I am wondering whether there is a quantifiable definition of superionicity? The molecular dynamics map does show the protons are all over the place, but I do not believe we should eyeball a superionic state.

Reply: There are different quantifiable definitions of ‘superionic phase’. In simulation works, superionic states are usually characterized by a finite positive diffusion rate of mobile atoms (H atoms in our case), while other atoms (Mg and O atoms in our case) vibrate at their equilibrium positions. If the diffusion rate of mobile atoms is larger than some cutoff value D , the state is defined as superionic. For example, in a paper about superionic BCC ice [*Phys. Rev. Lett.* **117**, 1–5 (2016)], the superionic regime is defined by $D_O = 0$ and $D_H > 0$ (D_O and D_H are diffusion coefficients of O and H atoms). In another work about superionic ice [*Nat. Phys.* **17**, 1228–1232 (2021)], the superionic transition temperatures are defined by $D_H = 10^{-8}$ m²/s. In a recent work about superionic helium–water compounds [*Nat. Phys.* **15**, 1065–1070 (2019)], the authors use a cutoff around $D = 10^{-9}$ m²/s ($D_H = 2.1 \times 10^{-9}$ m²/s and $D_{He} = 3.3 \times 10^{-9}$ m²/s are defined as ‘superionic’ in their work).

In this work, we calculate the diffusion coefficients from the slope of mean squared

displacement (MSD): $D = \text{MSD}/6t$, and define the superionic regime as $D_H > 10^{-9} \text{ m}^2/\text{s}$. Here are some diffusion rates from our simulations: at 600 GPa and 6000 K, $D_H = 1.11 \times 10^{-8} \text{ m}^2/\text{s}$ in $\text{Mg}_2\text{O}_3\text{H}_2$. $D_H = 9.30 \times 10^{-8} \text{ m}^2/\text{s}$ in MgO_3H_4 , $D_H = 1.37 \times 10^{-7} \text{ m}^2/\text{s}$ in MgO_4H_6 . More data can be found in Table S2. We've added this part to supplemental information.

(2) *What is the abundance of MgO and H2O in the interior of the planets? There must be a reasonable amount of both for the current prediction to be relevant.*

Reply: The abundance of different compositions in planets are still an open question. The ice to rock ratio in interior models of Uranus and Neptune is degenerated, varying from rock dominated to ice dominated models [*Philos. Trans. R. Soc. A Math. Phys. Eng. Sci.* **378**, (2020)]. Based on planet formation models, abundances of rocks and ices are expected to be similar in mass exterior to the ice-line (water condensation line). Interior models of planets suggest that rocky composition mainly consist of Mg, Si and Fe compounds [*Science* **214**, 145–149 (1981)], where MgO and SiO_2 are the most basic compounds. From the abundance of chemical elements in the universe [*Rev. Mod. Phys.* **47**, 877–976 (1975)], we can see that the abundance of Mg, Si and Fe are very similar. MgO is also the main topic in some recent studies [*Nat. Astron.* **5**, 815–821 (2021); *Nat. Astron.* **5**, 744–745 (2021)] about planet models. As a result, we can safely assume that MgO as one of the main components of the rocky core. H_2O is the most abundant ice, and the first to condense in the protoplanetary disk, thus it is expected to be main components of “hot ice” layer. We've added this part to the manuscript.

(3) *The predicted compounds contains a large amount of hydrogen, and therefore quantum effects are expected to be significant. Would the VASP MD and machine learning version of AIMD still be adequate to describe the behaviors of these compounds, in particular the dynamics of hydrogen? This part of simulation is perhaps more appropriate done with other methods such as path integral molecular dynamics. The authors may comment or justify their choice of method.*

Reply: We use path integral molecular dynamics (PIMD) implemented in i-PI [*Comput. Phys. Commun.* **185**, 1019–1026 (2014)] package, together with our machine learning force field to evaluate the nuclear quantum effects (NQEs). For comparison, we run a PIMD simulation with 16 beads and a classical MD (CMD) for MgO_4H_6 at 5000 K, 400 GPa. We compare their mean square displacement (MSD) and radial distribution function (RDF), as is shown in Figure R3. We find that the difference between PIMD and CMD is neglectable in this case. NQEs are often considered to be significant at relatively low temperature. Since our paper focus on the temperature condition of planetary interior ($10^3\sim 10^4\text{k}$), we consider NQEs can be neglected in our case. We've added this part to supplemental information.

Figure R3 | (a) Mean square displacement of CMD and PIMD. (b)-(d) Radial distribution function between O-H, H-H and O-O. The RDF is averaged over time.

Main modifications (marked with blue color in the revised manuscript):

1. We add the Mg-O polyhedral to the crystal structure picture in Fig. 1. (c)-(e).
2. The explanation of κ and σ has been added to Fig. 5.

3. We use green spheres to highlight the trajectory of a single H atom in Fig. 2.
4. Add some discussion about superionic behaviors and melting point, and add Fig. 3 (R2) into manuscript.
5. Add discussion about nuclear quantum effects into supplemental information and add Fig. S11 (R3).
6. We add some discussions on Uranus' low luminosity and refer to relevant works.
7. Change the caption of Fig. 6 to "Suggested interior models for ..."
8. Add explanation about "i-th flux" into the manuscript.
9. MgO_3H_4 is stable above 600 GPa, not 700 GPa. We correct this mistake.
10. We update the electronic conductivity data in the manuscript and Table S2.
11. We add definition of superionic phase and discussion about electronic conductivity to supplemental information.
12. We add discussion about abundance of H_2O and MgO to the manuscript.
13. Fix several typo errors.

References:

1. Hernandez, J. A. & Caracas, R. Superionic-Superionic Phase Transitions in Body-Centered Cubic H_2O Ice. *Phys. Rev. Lett.* **117**, 1–5 (2016).
2. Redmer, R., Mattsson, T. R., Nettelmann, N. & French, M. The phase diagram of water and the magnetic fields of Uranus and Neptune. *Icarus* **211**, 798–803 (2011).
3. Pickard, C. J., Martinez-Canales, M. & Needs, R. J. Decomposition and terapascal phases of water ice. *Phys. Rev. Lett.* **110**, 1–5 (2013).
4. Li, H.-F., Oganov, A. R., Cui, H., Zhou, X.-F., Dong, X. & Wang, H.-T. Ultrahigh-Pressure Magnesium Hydrosilicates as Reservoirs of Water in Early Earth. *Phys. Rev. Lett.* **128**, 035703 (2022).
5. Gao, H., Liu, C., Shi, J., Pan, S., Huang, T., Lu, X. & Wang, H. Superionic Silica-Water and Silica-Hydrogen Compounds in the Deep Interiors of Uranus and Neptune. *Phys. Rev. Lett.* **128**, 35702 (2022).

6. Cheng, B., Bethkenhagen, M., Pickard, C. J. & Hamel, S. Phase behaviours of superionic water at planetary conditions. *Nat. Phys.* **17**, 1228–1232 (2021).
7. Fortney, J. J. & Nettelmann, N. The interior structure, composition, and evolution of giant planets. *Space Sci. Rev.* **152**, 423–447 (2010).
8. Vazan, A. & Helled, R. Explaining the low luminosity of Uranus: A self-consistent thermal and structural evolution. *Astron. Astrophys.* **633**, 1–10 (2020).
9. Scheibe, L., Nettelmann, N. & Redmer, R. Thermal evolution of Uranus and Neptune: II. Deep thermal boundary layer. *Astron. Astrophys.* **650**, 1–12 (2021).
10. Kim, T., Chariton, S., Prakapenka, V., Pakhomova, A., Liermann, H. P., Liu, Z., Speziale, S., Shim, S. H. & Lee, Y. Atomic-scale mixing between MgO and H₂O in the deep interiors of water-rich planets. *Nat. Astron.* **5**, 815–821 (2021).
11. Nettelmann, N., Helled, R., Fortney, J. J. & Redmer, R. New indication for a dichotomy in the interior structure of Uranus and Neptune from the application of modified shape and rotation data. *Planet. Space Sci.* **77**, 143–151 (2013).
12. Helled, R., Anderson, J. D., Podolak, M. & Schubert, G. Interior models of Uranus and Neptune. *Astrophys. J.* **726**, 15 (2011).
13. Liu, C., Gao, H., Wang, Y., Needs, R. J., Pickard, C. J., Sun, J., Wang, H. T. & Xing, D. Multiple superionic states in helium–water compounds. *Nat. Phys.* **15**, 1065–1070 (2019).
14. Teanby, N. A., Irwin, P. G. J., Moses, J. I. & Helled, R. Neptune and Uranus: Ice or rock giants. *Philos. Trans. R. Soc. A Math. Phys. Eng. Sci.* **378**, (2020).
15. Hubbard, W. B. Interiors of the Giant Planets. *Science* **214**, 145–149 (1981).
16. Trimble, V. The origin and abundances of the chemical elements. *Rev. Mod. Phys.* **47**, 877–976 (1975).
17. Nettelmann, N. Stardust in the deep interior of low-mass planets. *Nat. Astron.* **5**, 744–745 (2021).

18. Ceriotti, M., More, J. & Manolopoulos, D. E. I-PI: A Python interface for ab initio path integral molecular dynamics simulations. *Comput. Phys. Commun.* **185**, 1019–1026 (2014).

REVIEWER COMMENTS

Reviewer #1 (Remarks to the Author):

The authors seems to address all of my comments and I suggest it to be published in its new revision.

Reviewer #3 (Remarks to the Author):

The authors have done a good job improving the manuscript and clarified my questions. Again, I think suggesting that water may be stored in MgO rock in the deep interiors of Earth to Neptune mass planets is interesting and justifies a publication in Nature Communicaitons. I am happy to recommend acceptance of this manuscript.